# Recent Advances in Enhancing the Therapeutic Index of PARP Inhibitors in Breast Cancer

**DOI:** 10.3390/cancers13164132

**Published:** 2021-08-17

**Authors:** Camille Franchet, Jean-Sébastien Hoffmann, Florence Dalenc

**Affiliations:** 1Laboratoire de Pathologie and Institut Claudius Regaud, Institut Universitaire du Cancer de Toulouse-Oncopole, 1 Av. Irène Joliot-Curie, 31100 Toulouse, France; Franchet.Camille@iuct-oncopole.fr; 2Laboratoire d’Excellence Toulouse Cancer (TOUCAN), Laboratoire de Pathologie, Institut Universitaire du Cancer-Toulouse, 31037 Toulouse, France; jean-sebastien.hoffmann@inserm.fr; 3Institut Claudius Regaud, Institut Universitaire du Cancer de Toulouse-Oncopole, 1 Av. Irène Joliot-Curie, 31100 Toulouse, France

**Keywords:** breast cancer, PARP inhibitor, homologous recombination deficiency, resistance

## Abstract

**Simple Summary:**

Two to three percent of breast cancer patients harbor germline mutation of either *BRCA*1 or *BRCA*2 genes. Their tumor cells are deficient in homologous recombination, a BRCA-dependent DNA repair machinery. These deficient cells survive thanks to the PARP-mediated alternative pathway. Therefore, PARP inhibitors have already shown some level of efficiency in the treatment of metastatic breast cancer patients. Unfortunately, some tumor cells inevitably resist PARP inhibitors by different mechanisms. In this review, we (i) present the notion of homologous recombination deficiency and its evaluation methods, (ii) detail the PARP inhibitor clinical trials in breast cancer, (iii) briefly describe the mechanisms to PARP inhibitors resistance, and (iv) discuss some strategies currently under evaluation to enhance the therapeutic index of PARP inhibitors in breast cancer.

**Abstract:**

As poly-(ADP)-ribose polymerase (PARP) inhibition is synthetic lethal with the deficiency of DNA double-strand (DSB) break repair by homologous recombination (HR), PARP inhibitors (PARPi) are currently used to treat breast cancers with mutated *BRCA*1/2 HR factors. Unfortunately, the increasingly high rate of PARPi resistance in clinical practice has dented initial hopes. Multiple resistance mechanisms and acquired vulnerabilities revealed in vitro might explain this setback. We describe the mechanisms and vulnerabilities involved, including newly identified modes of regulation of DSB repair that are now being tested in large cohorts of patients and discuss how they could lead to novel treatment strategies to improve the therapeutic index of PARPi.

## 1. Introduction

The poly(ADP-ribose) polymerases, PARP1 and PARP2, are known to transduce DNA damage signals through their high-affinity with single-stranded DNA breaks (SSBs). This is achieved by the activation of histone PARylation followed by chromatin decompaction and the loading of DNA repair factors. PARP inhibitors (PARPi) are particularly toxic in homologous recombination (HR)-deficient cells, notably in breast cancer (BC) cells mutated in HR genes *BRCA*. This is due to the trapping effect of the PARP enzymes on chromatin leading to a restricted accessibility to the repair factors and consequently to the accumulation of lesions that generate replication-coupled double-stranded DNA breaks (DSBs)/single-stranded DNA (ss DNA)-gaps, which need homology-directed repair by BRCA1/2 and RAD51.

PARPi are the first DNA repair inhibitor classes to be used for metastatic BC (mBC). At present, they are prescribed only as a single agent to patients with HR deficiency (HRD) due to germline *BRCA*1 or 2 (g*BRCA*) mutations. Proof of their effectiveness in HRD BC without g*BRCA* is still lacking.

The mechanisms of PARPi resistance are multiple and can be HR-dependent or -independent. Regardless of the mechanism, strategies to overcome resistance and prolong survival are paramount. These strategies must be better understood in order to define new strategies to overcome PARPi resistance.

## 2. HR Deficiency (HRD) and BRCA*ness* Phenotype

HRD is characterized by the lack of a proper HR in a cell, which is compensated by alternative non-homologous end-joining pathways. Thus, one of the major error-prone repair pathways of DNA DSBs is mediated by PARP. Therefore, this PARP-mediated rescue pathway is an excellent therapeutic target in cancer cells that frequently harbor HRD. We list the causes of HRD and briefly discuss methods for diagnosing HRD.

### 2.1. Causes of HRD

#### 2.1.1. Germline Mutations of Either BRCA1 or BRCA2

The most well-known HRD mechanism in BC is related to germline mutations in either *BRCA*1 (g*BRCA*1) or *BRCA*2 (g*BRCA*2) and concerns approximately 2 to 3% of all BCs [1]. In addition, BC patients harboring g*BRCA*1 are prone to triple-negative BC (TNBC), whereas g*BRCA*2 are associated with high-grade BCs, regardless of the phenotype. Thus, the prevalence of g*BRCA* BC is inversely correlated with age and depends on tumor immunophenotype and personal or family history of either breast or ovarian cancer. In a large study including 54,555 invasive BC patients, pathogenic mutations of either *BRCA*1 or *BRCA*2 were detected in 2.2% each. Indeed, the prevalence of *BRCA*1 or *BRCA*2 pathogenic mutations was, respectively, 0.9% and 2.1% in hormone receptor-positive (HR+)/human epidermal growth factor-receptor 2 (HER2)-negative BCs, 0.7% and 1.7% in HR+/HER2+ BCs, 1.8% and 1.3% in HR-negative/HER2+ BCs, and 6.7% and 2.7% in TNBCs [2].

#### 2.1.2. Germline Mutations of Additional DNA Damage Response (DDR)-Associated Genes

BCs harboring HRD without g*BRCA* mutations are considered to express what is known as BRCA*ness*. The BRCA*ness* phenotype might be due to several mechanisms. In two recent studies, several additional germline mutations in DDR genes in addition to g*BRCA* were associated with an increased risk of BC [3]. For instance, *PALB*2 codes for a protein involved in HR by its direct interaction with the recombinase RAD51 for strand invasion stimulation. Other genes carry a differential risk depending on the BC immunophenotype: *BARD*1, *RAD*51*C*, and *RAD*51*D* mutations are associated with a higher risk of ER-negative BCs, including TNBCs, whereas *ATM* and *CHK*2 mutations are associated with a higher risk of ER+ BC. Although germline mutations in *BRIP*1, *MUTYH*, or *OGG* have been reported, their association with BC risk remains unclear [4,5,6]).

#### 2.1.3. Somatic Mutations of BRCA1 or BRCA2

Only limited data are available on the prevalence and pathogenicity of somatic *BRCA* mutations in unselected BCs. In a TCGA BC study, somatic mutations of either *BRCA*1 or *BRCA*2 accounted for approximately one-third of *BRCA* mutations [7]. Similar ratios have been described in other studies [8,9,10]. Winter et al. reported nine somatic *BRCA* mutations out of 273 BCs, which included only two variants of unknown significance (VUS). On the other hand, in 122 BCs, Kwong et al. found 0.9% and 1.8% of somatic *BRCA*1 or *BRCA*2 mutations, respectively, exclusively composed of VUS [8,11]. Regarding the metastatic setting, the prevalence of either somatic *BRCA*1 or *BRCA*2 mutations was less than 2% in the SAFIR01/UNICANCER trial [12].

#### 2.1.4. Epigenetic Silencing of BRCA1, BRCA2 or RAD51C

*BRCA1* and *RAD51C* promoter hypermethylation has been considered as an epigenetic alteration leading to BC predisposition. Nevertheless, there are various pathogenesis hypotheses and conflicting results about its prevalence [13,14,15]. Furthermore, 4% of females in a Caucasian population were found to be carriers of low-level mosaic constitutional *BRCA1* methylation (4–10% of alleles) [16].

Evans et al. assessed the methylation of BRCA genes in 49 families with breast and ovarian cancer aggregation. Only two families (4.1%) harbored inherited *BRCA1* promoter hypermethylation with a methylation level of ~50% [13]. On the other hand, in a cohort of 613 patients harboring *BRCA*1/*BRCA*2 mutation-negative BCs, Hansmann et al. identified six (1%) and two (0.3%) patients with *BRCA*1 and *RAD*51*C* promoter hypermethylation, respectively [14]. Finally, Tabano et al. assessed *BRCA*1 and *RAD*51*C* hypermethylation in the blood of 89 high-risk BC patients and found neither g*BRCA* nor hypermethylation [15].

Concerning TNBC, higher frequencies of BRCA1 promotor hypermethylation ranging from 16% to 24% have been described [17,18]. Furthermore, the authors suggested that BRCA1 promoter hypermethylation may be twice as frequent as BRCA1 pathogenic variants in early-stage TNBC. Both studies also showed that *BRCA*-mutated TNBC and TNBC with BRCA1 promoter hypermethylation share common pathological and clinical features. Other studies have established a link between constitutional *BRCA*1 promotor hypermethylation and BC risk [19,20,21]. *BRCA*2 promoter hypermethylation has also been described in 5.3% of BCs [22].

Finally, while DNA methylation may be paramount for evaluating BCrisk, its assessment still suffers from the lack of harmonization in DNA sources, DNA methylation assessment techniques and measurement timing [23].

### 2.2. HRD Diagnosis

#### 2.2.1. Hereditary Breast and Ovarian Cancer Panel

In clinical practice, HRD is classically detected by Hereditary Breast and Ovarian Cancer (HBOC) panel sequencing. Genetic testing is now commonly recommended at the time of cancer diagnosis depending on factors such as age, personal/familial history of breast or ovarian cancer, and immunohistochemical subtype of BC. The panel not only contains *BRCA*1 and *BRCA*2, but also additional genes associated with moderate to high risk of breast and ovarian cancer such as *PALB*2, *TP*53, *PTEN*, *ATM*, and *CHK*2 [24].

#### 2.2.2. Genomic Signatures

Another way to detect HRD in tumors is based on the “genomic scars” concept. The error-prone backup repair pathways of DNA DSBs are associated with specific genomic alterations that can be considered as “signatures” of HRD [25]. Some of these alterations are related to chromosomal instability, such as large-scale transitions (LST), telomeric allelic imbalance (TAI), and loss of heterozygosity (LOH). Although their individual values remain suboptimal, these three chromosomal alterations are aggregated to form the genomic instability score, which is thresholded as a negative or positive HRD score in the commercially available Myriad Genetics test (MyChoice^®^ HRD) [26,27,28]. HRDetect is a BC-specific HRD signature based on whole genome sequencing data [29]. It relies on several genomic scars associated with BRCA*ness* in one score: point mutations, signature 3 according to Hollstein et al., microhomology, and DNA rearrangements and deletions [9,30,31,32]. It has already provided encouraging results in BC, but its predictive value of sensitivity to PARPi is still unclear. Interestingly, deletions flanking with microhomology seem to play an important role in this kind of HRD signature and could be considered as a footprint of theta-mediated end-joining [33,34,35]. Unlike HRDetect which requires extensive sequencing (WGS), the SigMA signature uses machine learning algorithms combined with targeted gene panels in order to detect HRD with no extensive sequencing [36]. However, it has not yet been validated in BC. Despite the promising results obtained with genomic scar signatures, they fail to perfectly predict the PARPi sensitivity of BC and suffer from a major drawback, in that they do not reflect fluctuations of HR in tumor evolution.

#### 2.2.3. Functional Analyses of HRD

To circumvent this issue, the most appealing method to assess HRD in real time is functional analyses. These approaches share a common principle: they quantify biomarkers involved in the HR pathway, BRCA1 itself or proteins downstream from BRCA1 and BRCA2.

RAD51 foci detection is a major biomarker of functional HR. These foci can be quantified in tumors with or without ex vivo treatment such as irradiation [37,38,39]. A significant decrease or a complete loss of RAD51 foci reflect HRD and confer PARPi sensitivity on BC tumor cells [39,40]. Importantly, functional HRD tests are currently under evaluation in clinical trials such as the RECAP test in the FUTURE trial (EudraCT 2018-002914-10), which is dedicated to the selection of mBC patients for talazoparib treatment [41].

## 3. Clinical Trials with PARPi

Currently, this class of drug includes five molecules: olaparib, talazoparib, rucaparib, niraparib, and veliparib. They differ from each other in terms of potency and cytotoxic effect owing to their different pharmacological mechanisms of action. PARPi act in two main ways: (i) as a DNA repair inhibitor through the inhibition of the enzymatic activity of PARP, and (ii) as a DNA damaging agent by trapping PARP at sites of DNA damage. Although olaparib and talazoparib share approximately the same PARylation catalytic level of inhibition, talazoparib is one-hundred times more potent than olaparib at stabilizing PARP-DNA complexes (PARP trapping) [42]. Preclinical studies suggest that PARP trapping on DNA may induce cancer cell death more efficiently than catalytic inhibition of PARP, but this strategy carries more haematological toxicity so it cannot be combined with chemotherapy (CT) [43,44].

### 3.1. PARPi in Monotherapy

#### 3.1.1. Patients with Germline BRCA Mutations

Two open multicenter phase 3 trials conducted with olaparib and talazoparib (OlympiAD and EMBRACA trial, respectively) enrolled 302 and 431 patients with HER2-negative mBC who had received ≤2 or ≤3 prior CT for stage IV, randomizing them 2:1 to receive PARPi or CT according to the investigator’s choice. Progression-free survival (PFS), which was the primary endpoint, was higher in the PARPi arm compared to CT in both trials: 7.0 vs. 4.2 months (HR 0.58; 95% CI 0.43–0.80; *p* < 0.001) in OlympiAD [45] and 8.6 vs. 5.6 months (HR 0.54; 95% CI 0.41–0.71; *p* < 0.001) in EMBRACA [46]. The median overall survival (OS) was not improved by PARPi [47,48]. However, neither trial was sufficiently powered to detect OS differences and the uncontrolled treatment crossover after discontinuation of the study drug might have been a confounding factor in the statistical analysis. BRAVO (NCT019005592) is a phase 3 randomized trial assessing the activity of niraparib compared to a single CT agent of the investigator’s choice in which results are awaited.

Concerning early BC, in the phase 2 BRE09-146 trial of the Hoosier Oncology Group, rucaparib failed to improve outcomes in terms of disease-free survival (DFS) at 2 years in 128 patients with TNBC or *gBRCA* mutations who received cisplatin with or without rucaparib for residual disease after anthracycline/taxane-based treatment [49]. Litton et al. reported a trial conducted in neoadjuvant therapy with talazoparib administered for 6 months in 20 patients with operable stage I–III, HER2-negative BC with a g*BRCA* mutation [48]. RCB-0 was 53% and RCB-0/I was 63% at the cost of essentially hematological toxicity managed by dose reduction and transfusions. This encouraging result was recently confirmed in an independent series of 61 patients by the same author (NEOTALA trial) during the ASCO 2021 meeting [50]. Importantly, adjuvant olaparib after completion of local treatment and (neo)adjuvant CT were recently associated with significantly longer survival free of invasive (HR 0.58; 95% CI 0.41–0.82; *p* < 0.001) or distant disease (HR 0.57; 95%CI 0.39–0.83; *p* = 0.02) compared to placebo in 1836 patients with high-risk, HER2-negative BC and g*BRCA1* or g*BRCA2* in the OlympiA trial [51]. Data in terms of OS are awaited.

#### 3.1.2. Patients without Germline BRCA Mutation but *BRCAness* Tumours

Recently, Tung and al. reported encouraging results in the phase 2 study TBCRC 048 with olaparib for patients with mBC and g*PALB*2 mutations (n = 11) with 82% of objective response rate (ORR), or somatic *BRCA*1/2 mutations (n = 16) with 51% of ORR [52]. These results suggest that patients with mBC might benefit from PARPi as much as g*BRCA* mutation carriers. Currently, patients with *BRCAness*, stage III, HER2-negative BC are being enrolled in the SUBITO trial, sponsored by the Netherlands Cancer Institute (NCT02810743) and are receiving olaparib for 1 year. In the phase II window study RIO (n = 43), rucaparib administered for 2 weeks in patients with TNBC failed to decrease Ki67. However, circulating tumor DNA (ctDNA) completely disappeared with rucaparib in patients with mutation-signature HR-deficient TNBC (69% of the patients) [53].

### 3.2. PARPi in Combination with Chemotherapy

As many cytotoxics cause damage to the DNA of cancer cells and because PARPi alter DNA repair mechanisms, some authors have hypothesized that PARPi could be used as chemosensitizers. Veliparib is considered to be the least cytotoxic PARPi as it acts through inhibition of PARP inhibition but does not trap PARP in the DNA.

The phase III randomized (2:1) BROCADE 3 trial compared the activity of carboplatin and paclitaxel with either veliparib or placebo in 513 g*BRCA*-mutated, HER2-negative mBC patients [54]. If CT was stopped, veliparib was continued as a single agent at full dose. Median PFS was improved in the veliparib group: 14.5 vs. 12.6 months (HR 0.71; 95% CI 0.57–0.88, *p* = 0.0016). Interestingly, the curves did not appear to separate until after about 12 months of treatment coinciding with the maintenance phase, suggesting no benefit with the combination and raising the question of the clinical utility of veliparib in maintenance after induction CT. The SAFIR-02 study (NCT02299999) also posed the question of the clinical benefit of a tailored therapy including a PARPi (olaparib) after induction CT in patients with *BRCAness* mBC. Results are awaited.

The randomized, double-blind, phase III *BrighTNess* trial was developed to determine whether the results of the Y-SPY2 trial [55] could be confirmed and, if so, to determine the contribution of veliparib to the increase in the pathological complete response (pCR) rate [56]. This trial included 634 women with stage II/III TNBC with (15%) or without a g*BRCA* mutation. They receive paclitaxel alone (arm A) or paclitaxel + carboplatin (arm B) or paclitaxel + carboplatin + veliparib (arm C). The pCR rate was greater in the arm C than in the arm A (53% vs. 31%, *p* < 0.0001). However, the addition of veliparib did not increase pCR compared to the association paclitaxel + carboplatin. *BrighTNess* corroborated previous results on the benefit of adding carboplatin as neoadjuvant in early TNBC in terms of pCR but did not show the benefit of adding a PARPi, including in the population with a *gBRCA* mutation. In the randomized phase II GeparOLA trial, olaparib failed to increase the rate of pCR compared to platinum in patients with HER2-negative BC with or without a *gBRCA* mutation [57]. The phase I/II PARTNER (NCT 03150576) trial, whose design is similar to the *BrighTNess* trial, is ongoing with olaparib. It is designed to demonstrate an increase in pCR with olaparib in addition to CT in patients with TNBC with or without a *gBRCA* mutation.

### 3.3. PARPi in Combination with Other Agents

BC associated with a *gBRCA* mutation carry specific molecular alterations that include a high tumor mutational burden. Moreover, olaparib has been shown to stimulate PD-L1 expression in *gBRCA*-mutated tumor cells inhibiting lymphocyte cytotoxicity [58]. More recently, Ding et al. have shown that olaparib elicits an antitumour response in BRCA1-deficient ovarian tumors in mice. This immune response is mediated by both intratumoural and peripheral effector CD4+ and CD8+ T cells through a stimulator of interferon genes (STING)-dependent signal [59]. These data support the combination of PARPi and immunological checkpoint inhibitors. The phase I-II MEDIOLA trial reported an ORR of 63.3% (95% CI 48.9–80.1) in 34 patients with *gBRCA* mutation and mBC with olaparib plus durvalumab [60]. In the phase I-II KEYNOTE 162 trial, the association niraparib and pembrolizumab provided an ORR of 55% in 55 patients with mTNBC with or without a *gBRCA* mutation and unselected by PD-L1 positivity [61]. Finally, several studies combining PARPi and anti-PD(L)1 or radiotherapy or PI3K inhibitors or conjugate drug antibody are ongoing (Table 1).

## 4. Mechanisms of PARPi Resistance

Although ~40% of BC exhibit primary resistance, little is known about this phenomenon. Conversely, secondary resistance has been extensively studied both in vivo and in vitro. However, little is known about these mechanisms in clinical practice. In this chapter, we describe resistance mechanisms to PARPi in BC, with a special emphasis on mechanisms encountered in clinical practice. PARPi resistance mechanisms are frequently split in two categories: HR-dependent and HR-independent mechanisms (Figure 1).

### 4.1. HR-Dependent Mechanisms

#### 4.1.1. Reversion Mutations

One of the best described mechanisms both in vivo and in clinical practice is the reversion mutation of *BRCA*. Several studies have identified tumor-specific *BRCA*1 or *BRCA*2 secondary mutations in PARPi-resistant BC cells, both in tissue and in ctDNA [62,63,64]. In a meta-analysis by Tobalina et al., both ctDNA and tumor DNA sequencing data of 27 BC that progressed under platinum or PARPi treatment were analyzed in order to identify reversion mutations [65]. Fifteen reversion mutations were discovered in 10 patients. There was no significant difference between patients with either the g*BRCA*1 or g*BRCA2* mutation. In most cases, reversions involved small insertions or deletions that restored the open-reading frame. Deletions accounted for most secondary mutations. These deletions could be explained by error-prone DNA repair pathways such as microhomology-mediated end-joining (MMEJ). Interestingly, the authors identified an MMEJ signature in 9 out of 15 reversion mutations, preferentially in the ctDNA of PARPi-treated patients. In such cases of reversion mutations, Waks et al. showed in a small cohort of eight BC that HR was successfully restored, as RAD51 foci were absent before and detectable again after PARPi or platinum-based treatment [66]. Taken together, these results demonstrate that reversion mutations occur frequently under PARPi treatment and eventually restore HR in tumour cells.

#### 4.1.2. Loss of DNA End Resection Inhibition

DNA end resection leads to a 3′ ss-DNA tail that can invade the homologous DNA strand on the sister chromatid, thus its crucial role in ensuring proper HR. The inhibition of DNA end resection mediated by the Shieldin complex—REV7, SHLD1-3—under the control of 53BP1 and RIF1 directs DNA double-strand (ds)-break repair towards non-homologous end-joining (c-NHEJ). In fact, 53BP1 depletion is known to restore BRCA1-independent HR in BRCA1-deficient tumor cells and to confer resistance to PARPi [40,67,68,69]. It has also been recently described in a few cases of BRCA1-deficient BC treated by PARPi [66]. Interestingly, this resistance mechanism could be overcome by cyclin-dependent kinase inhibitor treatment in g*BRCA* HR+ BC [70]. Indeed, the latter seems to restore G1 cell cycle arrest and to promote c-NHEJ targeted by PARPi.

As 53BP1-dependent DNA repair is mediated by the Shieldin complex, depletion of the Shieldin complex has also been associated with HR reactivation and PARPi resistance [71,72]. Collectively, these results show that DNA end resection inhibition leads to PARPi resistance in BRCA1-deficient tumor cells through the restoration of BRCA1-independent HR. PALB2 and ubiquitin E3 Ligase mediate HR in these situations [73,74,75].

### 4.2. HR-Independent Mechanisms

Besides HR-dependent mechanisms of PARPi resistance, numerous HR-independent mechanisms have also been hypothesized, although few have been supported by clinical evidence so far.

#### 4.2.1. Upregulation of Drug Efflux Pumps

As widely described for all kinds of CT, upregulation of ABC transporters might play a role in PARPi treatment [76]. Christie et al. posited that upregulation of *ABCB1* through transcriptional fusion—leading to increased multidrug resistance protein (MDR1) in tumor cells—should be associated with the number of previous MDR1-substrate CT lines. Indeed, paclitaxel is frequently used for BC treatment and could precondition PARPi resistance [77].

#### 4.2.2. PARP Activity Alteration

As the cytotoxic effect of PARPi functions through PARP trapping on DNA damage, trapping-diminishing *PARP*1 mutations are thought to result in PARPi resistance [78]. Zandarashvili et al. recently showed that trapping effect of PARPi is dependent of its capability to destabilize the helical domain of PARP1 [79]. The authors described three types of PARPi related to this PARP1 allostery: type I, allosteric pro-retention on DNA; type II, non-allosteric; and type III, allosteric pro-release from DNA. Therefore, only type I PARPi could be subject to resistance due to trapping-diminishing *PARP*1 mutations.

Interestingly, this effect not only depends on the PARPi molecule, but on the type and location of *BRCA* mutation too. In fact, *BRCA*1-mutated tumor cells that retain some BRCA function may be especially prone to this mechanism of PARPi-resistance.

Poly (ADP-ribose) glycohydrolase (PARG) counteracts PARP1 activity by degrading PAR chains. Therefore, PARG works in the same direction as PARPi. In fact, PARG depletion in BC has been shown to result in PARPi resistant in mouse models [80]. PAR accumulation caused by PARG depletion could prevent PARP trapping and promote PARPi resistance. Nevertheless, this hypothesis remains to be confirmed clinically.

#### 4.2.3. Increased Stabilization of Replication Forks

As BRCA1 and BRCA2 ensure the stability of stalled replication forks, their depletion leads to a dramatic increase in ds-DNA breaks in HR-deficient cells. Therefore, PARPi resistance could also be achieved by reducing the number of ds-DNA breaks in tumor cells. In this respect, mechanisms that rescue stalled replication fork stabilization might overcome BRCA deficiency. Fork collapses in BRCA-deficient cells are mediated by three pathways: the SMARCAL1-MRE11 axis, the EZH2-MUS81 axis, and the PTIP-MRE11 axis [81]. Therefore, loss of either PTIP or EZH2 has been proposed as a potential PARPi resistance mechanism [82,83]. Interestingly, PTIP depletion seems to lead to PARPi resistance irrespective of the BRCA1 or BRCA2 inactivation, whereas EZH2 depletion would only lead to resistance in BRCA2-deficient cells.

## 5. Novel Strategies for Future Combination Therapies to Overcome Resistance to PARP Inhibitors in Breast Cancers

Given the limited clinical efficacy of PARPi, alternative approaches to overcome resistance are now needed by targeting the acquired vulnerabilities of PARPi-resistant tumors. We now review strategies that may be used in BC to overcome these limitations.

### 5.1. Inhibition of the ATR/CHK1/WEE1 Pathway to Restore PARPi Sensitivity

DNA damage-induced checkpoints are known to coordinate DDR and cell cycle control by avoiding the generation of excessive DNA damage during DNA replication and mitosis and allowing sufficient time for repair. The PI3K-related protein kinases ATR and ATM play a central role in this coordination and guarantee the integrity of the genome [84]. Activation of ATM by DSBs leads to initiation of the HR-pathway machinery through the activation of BRCA1, BRCA2, and PALB2 as well as activation of the CHK2 kinase, which in turn activates p53. ATR and its downstream effector CHK1 are primarily activated by the accumulation of ssDNA coated with phosphorylated RPA (pRPA) following stalled or collapsed DNA replication forks [85]. In addition, activation of ATR/CHK1 leads to the activation of WEE1 and inhibition of CDC25A and CDC25B [86]. Thus, activation of the ATR/CHK1/WEE1 pathway responds to replication stress and extends the S/G2 phases of the cell cycle. Treatment with PARPi, which induce a strong replicative stress, leads to the activation of this ATR/CHK1 pathway [87]. Several studies using different cancer models suggest that PARPi-resistant tumors are strongly dependent on the ATR/CHK1/WEE1 pathway, and that inhibition of these kinases may restore PARPi sensitivity (Figure 2). This is the case (a) in PARPi-resistant BRCA-deficient cancer cells, where ATR inhibition inhibits the restored HR and affects fork protection, thereby resensitizing these cells to PARPi [88], and (b) in cancer preclinical models resistant to PARPi, where CHK1 inhibitor abolishes the resistance by destabilizing replication forks [89]. In this respect, an ongoing Phase 1-trial is currently assessing the efficacy of the association Olaparib-CHK1 inhibitor as a possible treatment for high-grade serous ovarian or fallopian tube cancer (NCT03057145). Nevertheless, to the best of our knowledge, this association has not been tested in BC yet.

Besides the repair of single-strand DNA breaks, PARP is also known to be part of ligation mechanism of the Okazaki fragments within replication forks and contributes to fork stability together with the ATR/CHK1/WEE1 pathway. In cancers with strong RS such as breast cancer with the *BRCA*1/2 mutation, targeting these two different fork stabilizing mechanisms by the combination of PARPi and ATRi would lead to increased DNA ds-breaks and tumor cell death. The use of ATRi would be notably beneficial in PARPi-resistant cells with restored fork protection.

Combining PARPi with ATR inhibitors (ATRi) therefore holds great promise for overcoming PARPi resistance in tumors with restored HR or restored fork protection. This was already observed in ovarian cancer models and needs to be tested in BC. Potent and selective ATRi inhibitors (ATRi) such as AZD673841 and M662042 are in phase I/II clinical trials (clinicaltrials.gov accessed date: 8 May 2021).

### 5.2. Potential Use of POLQ Inhibitors for PARPi-Resistant Breast Cancers

Besides the c-NHEJ pathway known to repair most DSBs by a ligation mechanism of DNA ends without extensive processing [90], HR is the preferential DSB repair pathway in the S and G2 phases of the cell cycle, when a sister chromatid is available [91]. A third pathway, termed polymerase theta-mediated end joining (TMEJ) due to the requirement of the specialized DNA polymerase θ (Polθ) encoded by the *POLQ* gene [92], operates on a common resected HR intermediate. TMEJ is initiated by PARP1 and Polθ recruitments to these resected DNA-ends. Polθ can then form dimers, which facilitate the stabilization of synapsed intermediates and perform a bidirectional scanning initiated from the 3′ termini to detect internal microhomologies, which can be annealed, thus generating 3′ flaps. Polθ itself can remove the 3′ flaps [93] and initiate the repair DNA synthesis with poor processivity and frequent aborted synthesis, resulting in a high rate of mutations (Figure 3) [94]. Importantly, the processing of broken DNA ends is essential to orientate the repair processes. In fact, DNA resection is known to be blocked by the chromatin-binding protein 53BP1, which shields DNA ends from nucleases through its effectors Shieldin and Rif1 [95]. Considering the competition between HR and TMEJ on resected DNA ends at DSB, it was predictable that HR-deficient cancer cells could depend on TMEJ for their repair. Indeed, depletion of Polθ in an HR-deficient background affected cell viability and a synthetic lethal relationship between Polθ and HR factors was revealed [96]. Importantly, depletion of Polθ was shown to further increase the sensitivity of HR-deficient cells to PARPi, and combining PARP inhibition with Polθ inhibition also improved the antiproliferative effects [96,97]. Furthermore, the *BRCA* mutations observed in PARPi-resistant cells often exhibit a TMEJ signature [98], suggesting that Polθ itself might contribute to the acquisition of PARPi resistance. It has been reported that resistance to PARPi can also occur through loss of 53BP1 or its effectors Shieldin, which render HR-deficient cells entirely dependent on Polθ [99,100]. Consequently, the use of Polθ inhibitors in combination with PARPi, or as a second-line therapy, may abolish or delay resistance acquisition in some selected patients and extend the response to PARPi [97,101].

Importantly, two recent works reported the discovery and characterization of the first-in-class specific inhibitors of Polθ [72,97], providing an exciting opportunity to explore their utility to treat a sub-population of patients with BRCA1/2 mutations who show resistance to PARP-inhibitor therapy.

### 5.3. Inhibition of Chromatin Remodelers Combined with PARPi

The chromatin environment of the genome is an obstacle to efficient DNA repair pathways, as it hinders the access and processing of different types of DNA breaks and lesions by repair factors, and requires nucleosome-remodeling events in response to genomic signals and stresses. Chromatin decompaction and loading of repair factors in response to ssDNA breaks is facilitated by the action of PARP proteins and the recruitment of PAR-binding effector proteins. The PAR-binding chromatin remodeler ALC1, which was recently identified as an actor of PARPi toxicity in HR-deficient cells through the accumulation of PARP trapping and stalled replication forks [102], is a novel potential therapeutic target in HR-deficient breast cancers.

## 6. Conclusions

A significant proportion of HER2-negative BC exhibit HR DNA repair pathway deficiency. In clinical practice, gBRCA1/2 mutation is the only mechanism of HR deficiency currently assessed in mBC leading to PARPi indication. Patients harboring those tumors may take advantage of a wide variety of new treatments. Regarding these emergent therapies, PARPi appears to be the first leading to an increase of PFS in mBC. Unfortunately, primary or acquired resistance to PARPi may naturally occur. Herein, we describe all the current clinical evidence supporting PARPi in BC and its mechanisms of resistance. Actually, the introduction of PARPi in clinical practice greatly improved the understanding of biological mechanisms underlying resistance and provide several tracks to overcome it. Indeed, we stressed the prolific nature of clinical research in this field. On the one hand, several ongoing trials assess the efficacy of PARPi associated with other treatments. On the other hand, numerous new DDR targets inhibitors are developed simultaneously, and we expect them to show positive results in the near future.

The challenges are now to trigger the development of novel non-cross-resistant therapies and support the optimization of its prescription sequence. In addition, we should more accurately identify non-BRCA germline-mutated patients that could benefit from PARPi or others, especially those harboring HRD tumors.

## Figures and Tables

**Figure 1 cancers-13-04132-f001:**
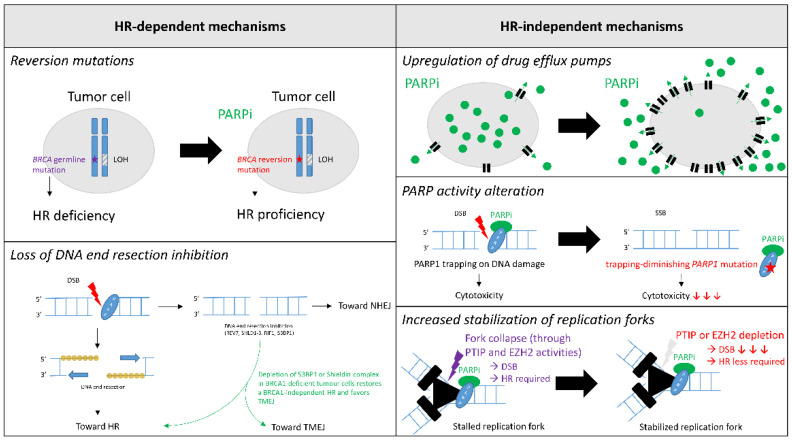
Synthetic representation of PARP inhibitors resistance mechanisms. Abbreviation: DSB = Double strand break; HR = Homologous recombination; LOH = Loss of heterozygosity; NHEJ = Non-homologous end-joining; PARP = Poly(ADP-ribose) polymerase; PARPi = PARP inhibitors; SSB = Single strand break; TMEJ = Theta-mediated end-joining.

**Figure 2 cancers-13-04132-f002:**
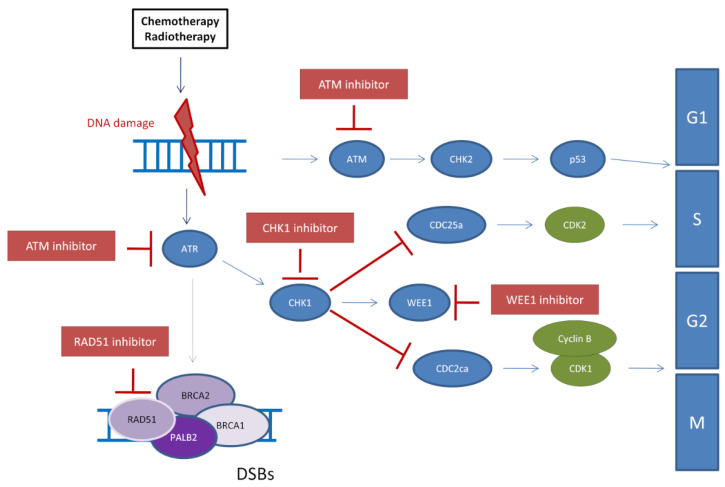
Therapeutic strategies to target DDR. Abbreviation: DSBs = Double-strand breaks.

**Figure 3 cancers-13-04132-f003:**
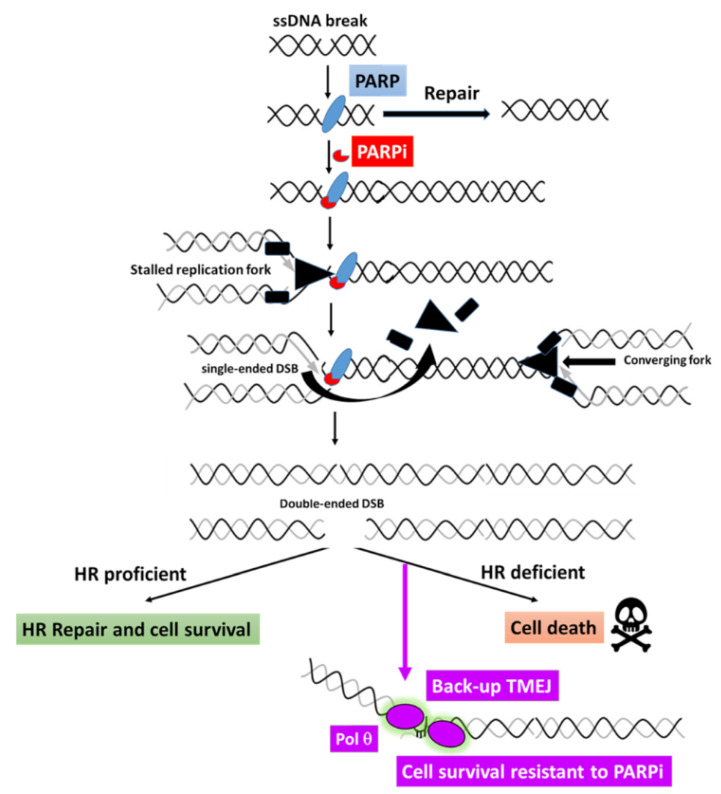
Rationale for targeting theta-mediated end joining (TMEJ) in PARPi-resistant breast cancers. The poly(ADP-ribose) polymerase (PARP) is responsible for the repair of single-stranded DNA (ssDNA) breaks. The PARP inhibitors inhibit the catalytic activity of PARP and can trap PARP proteins on the DNA. These bulky complexes impede the progression of DNA replication forks, which stall, leads to unloading of the replicative helicase and DNA polymerases, and creates a single-ended DSB. When a converging fork is coming, a double-ended DSB is generated. This DSB can be resolved by a functional HR machinery, leading to cell survival. When HR is defective, the damage persists, leading to cell death. One mechanism explaining primary or secondary resistance to PARPi relies on the activation of the alternative Polθ-mediated End-Joining (TMEJ) DSB repair pathway which can rescue HR defect. Abbreviations: HR = homologous recombination; ssDNA = single-stranded DNA.

**Table 1 cancers-13-04132-t001:** Main ongoing trials in BC combining PARPi and other anti-tumour therapies.

Clinical Trial	Phase	Patient Population	PARPi	Other Anti-Tumoural Therapy
**DORA** NCT03167619	II	mTNBC after carboplatine	olaparib in maintenance	durvalumab
NCT03801369	II	mTNBC	olaparib	durvalumab
**DOLAF** NCT04053322	II	ER+/HER2-negative mBC with gBRCA mutation or HRD+	olaparib	durvalumab and fulvestrant
NCT02849496	II	HER2-negative and HRD+ mBC	olaparib	atezolizumab
NCT02484404	I-II	mTNBC	olaparib	durvalumab and cediranib (anti-angiogenic)
NCT03025035	II	gBRCA mutation mBC	olaparib	pembrolizumab
NCT03598257	II	inflammatory BC	olaparib	radiotherapy
**UNITY** NCT03945721	I	early TNBC with residual disease after NACT	niraparib	radiotherapy
NCT03542175	I	early TNBC with residual disease after NACT	rucaparib	radiotherapy
NCT01623349	I	mTNBC	olaparib	BKM120 ou BYL719
NCT04586335	I	HRD+ and/or PI3K mutation solid cancer	olaparib	CYH33
NCT04039230	I-II	mTNBC	talazoparib	Sacituzumab-govitecan
NCT03901469	II	mTNBC without gBRCA mutation	talazoparib	ZEN003694

Abbreviations: NACT = neoadjuvant chemotherapy, mTNBC = metastatic triple-negative breast cancer; gBRCA mutation = germline *BRCA* mutation.

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
