# Peer review of "Recent Advances in Enhancing the Therapeutic Index of PARP Inhibitors in Breast Cancer"

_cancers, 2021, doi:10.3390/cancers13164132_

Round 1
Reviewer 1 Report
This manuscript By Dr. Dalenc and his colleagues summarized the current knowledge of PARP inhibitors in Breast Cancer. The article is well written and provides thorough reviews on this important topic. A couple of minor points, however, should be considered:
- For PARPi in combination with chemotherapy section, authors need to include clinical trial of PARPi with other inhibitors (for example PARPi with CHK1 inhibitor: NCT03057145)
- It would be helpful to the readers if the authors can generate a figure summarizing the mechanisms of PARPi resistance
- It will add value to the manuscript if the authors could include the conclusion at the end of manuscript and make comments on the future directions.
Author Response
1. For PARPi in combination with chemotherapy section, authors need to include clinical trial of PARPi with other inhibitors (for example PARPi with CHK1 inhibitor: NCT03057145)
We warmly thank the reviewer for this comment. Actually, this ongoing clinical trial only include patients with high-grade serous ovarian or fallopian tube cancer. However, we have cited this trial in the section 5.1 in the paragraph dedicated to ATR/CHK1/WEE1 pathway.
2. It would be helpful to the readers if the authors can generate a figure summarizing the mechanisms of PARPi resistance
As required, an additional figure (Figure 1) has been included in the manuscript.
3. It will add value to the manuscript if the authors could include the conclusion at the end of manuscript and make comments on the future directions.
A conclusion has been added to the manuscript including comments on the future directions.
Reviewer 2 Report
The manuscript described the therapeutic impact of PARP inhibitors (PARPi) and discuss the potential treatments overcoming PARPi -resistance in breast cancer. The reviewer can appreciate the authors’ focus point. However, the authors are asked to address the following issues before consideration of publishing the manuscript in Cancers.
1) Tutt , et al. reported that adjuvant olaparib after completion of local treatment and neoadjuvant or adjuvant chemotherapy was associated with significantly longer survival free of invasive or distant disease than was placebo in breast cancer patients with germline BRCA mutation (NEJM 2021; 384:2394). To reflect the latest evidence, the authors are highly asked to cite this paper and describe the results in OlympiA study in 3.1.1.
2) PARPi combining with immunotherapy were summarized in Table 1. The authors described the rationale of this combination focused on neo-antigen and PD-L1 upregulation. The previous studies demonstrated that DNA damaging agents may actually promote immunogenic cell death, alter the inflammatory milieu of the tumor microenvironment and/or stimulate neoantigen production, thereby activating an antitumor immune response (Brown J, et al. British J Cancer 2018; 118, 312–324). Recently, Ding, et al. showed that PARP inhibition elicits STING-dependent antitumor immunity in BRCA-deficient tumor (Cell Reports 2018ï¼›25:2972–2980). The authors are asked to describe the mode-of- action of olaparib in immune-oncology in 3.3.
4) Clinical PARP inhibitors including olaparib, talazoparib, niraparib bind the same catalytic domain of PARP but exert the different clinical outcomes. Murai, et al. summarized the comparison of clinical PARPi based on the individual parameter (Annu Rev Cancer Biol 2019; 3:131). The authors are asked to compare cytotoxicity among these clinical PARP inhibitors in the section 3.
5) Recently, Zandarashvili reported that structural basis for allosteric PARPi retention on DNA break (Science 2020; 368: 46). PARP inhibitors were classified into pro-retention, mild and pro-release type by allosteric effects on PARP-1. In the section of 4.2.2, the authors are asked to discuss trapping-diminishing PARP1 mutation in related to this PARP allostery.
Author Response
1. Tutt , et al. reported that adjuvant olaparib after completion of local treatment and neoadjuvant or adjuvant chemotherapy was associated with significantly longer survival free of invasive or distant disease than was placebo in breast cancer patients with germline BRCA mutation (NEJM 2021; 384:2394). To reflect the latest evidence, the authors are highly asked to cite this paper and describe the results in OlympiA study in 3.1.1.
We apologize for this. In fact, the OlympiA study was already mentionned in the previous paragraph of the 3.1.1 section with the appropriate citation. Nevertheless, it was written « OlympiAD » and not « OlympiA ». This has been corrected in the revised version.
2. PARPi combining with immunotherapy were summarized in Table 1. The authors described the rationale of this combination focused on neo-antigen and PD-L1 upregulation. The previous studies demonstrated that DNA damaging agents may actually promote immunogenic cell death, alter the inflammatory milieu of the tumor microenvironment and/or stimulate neoantigen production, thereby activating an antitumor immune response (Brown J, et al. British J Cancer 2018; 118, 312–324). Recently, Ding, et al. showed that PARP inhibition elicits STING-dependent antitumor immunity in BRCA-deficient tumor (Cell Reports 2018ï¼›25:2972–2980). The authors are asked to describe the mode-of- action of olaparib in immune-oncology in 3.3.
This point has been added to our manuscript as requested by the reviewer.
3. Clinical PARP inhibitors including olaparib, talazoparib, niraparib bind the same catalytic domain of PARP but exert the different clinical outcomes. Murai, et al. summarized the comparison of clinical PARPi based on the individual parameter (Annu Rev Cancer Biol 2019; 3:131). The authors are asked to compare cytotoxicity among these clinical PARP inhibitors in the section 3.
We warmly thank the reviewer for this accurate advice. The cytotoxicities of both talazoparib and olaparib have been compared in the section 3 and we have added the proposed reference.
4. Recently, Zandarashvili reported that structural basis for allosteric PARPi retention on DNA break (Science 2020; 368: 46). PARP inhibitors were classified into pro-retention, mild and pro-release type by allosteric effects on PARP-1. In the section of 4.2.2, the authors are asked to discuss trapping-diminishing PARP1 mutation in related to this PARP allostery.
This notion has been added to the manuscript as requested by the reviewer, along with the proposed reference.
Round 2
Reviewer 2 Report
The authors have responded to the reviewer's comments correctly and the manuscript has been improved significantly.